# Bacterial Succession during Vermicomposting of Silver Wattle (*Acacia dealbata* Link)

**DOI:** 10.3390/microorganisms10010065

**Published:** 2021-12-29

**Authors:** Daniela Rosado, Marcos Pérez-Losada, Manuel Aira, Jorge Domínguez

**Affiliations:** 1CIBIO-InBIO, Centro de Investigação em Biodiversidade e Recursos Genéticos, Campus Agrário de Vairão, Vairão, Universidade do Porto, 4485-661 Porto, Portugal; mlosada323@gmail.com; 2Computational Biology Institute, The George Washington University, Washington, DC 20052, USA; 3Department of Biostatistics & Bioinformatics, Milken Institute School of Public Health, The George Washington University, Washington, DC 20052, USA; 4Grupo de Ecoloxía Animal (GEA), Universidade de Vigo, E-36310 Vigo, Spain; aira@uvigo.es (M.A.); jdguez@uvigo.es (J.D.)

**Keywords:** 16S rRNA, earthworms, metataxonomics, microbiome, vermicompost

## Abstract

Vermicomposting is the process of organic waste degradation through interactions between earthworms and microbes. A variety of organic wastes can be vermicomposted, producing a nutrient-rich final product that can be used as a soil biofertilizer. Giving the prolific invasive nature of the Australian silver wattle *Acacia dealbata* Link in Europe, it is important to find alternatives for its sustainable use. However, optimization of vermicomposting needs further comprehension of the fundamental microbial processes. Here, we characterized bacterial succession during the vermicomposting of silver wattle during 56 days using the earthworm species *Eisenia andrei*. We observed significant differences in α- and β-diversity between fresh silver wattle (day 0) and days 14 and 28, while the bacterial community seemed more stable between days 28 and 56. Accordingly, during the first 28 days, a higher number of taxa experienced significant changes in relative abundance. A microbiome core composed of 10 amplicon sequence variants was identified during the vermicomposting of silver wattle (days 14 to 56). Finally, predicted functional profiles of genes involved in cellulose metabolism, nitrification, and salicylic acid also changed significantly during vermicomposting. This study, hence, provides detailed insights of the bacterial succession occurring during vermicomposting of the silver wattle and the characteristics of its final product as a sustainable plant biofertilizer.

## 1. Introduction

Vermicomposting is the production of organic amendment via the degradation of organic waste through interactions between earthworms and microbes [1,2]. While earthworms ingest and process the organic waste (active phase), microbes degrade the organic matter processed by earthworms (maturation phase) during vermicomposting [1]. Additionally, earthworms have a direct impact on the microbial communities since they ingest microbes that can either be digested or released again into the environment [3]. The temporal changes of the microbial community composition and organic matter decay during vermicomposting are an example of heterotrophic ecological succession [4,5]. The microbial succession is driven by the quantity and quality of the available nutrients in the initial substrate such as organic carbon (e.g., [6]). During vermicomposting, the microbial diversity of the substrate can be modified by the earthworms through gut-associated processes [7,8]. In addition, the progression of microbial succession is characterized by the replacement of specific groups of bacteria by others, which facilitates substrate utilization and metabolization of the remaining nutrients during the maturation stage (e.g., [6]).

A great variety or organic waste can be effectively vermicomposted including industrial and agricultural wastes [1]. The final product (vermicompost) is rich in microorganisms and nutrients and can be used as a soil biofertilizer [9]. Vermicompost has been shown to promote plant growth due to the presence of growth regulating substances as well as its ability to mitigate or suppress plant diseases (reviewed in [9]). Thus, vermicomposting has the potential to convert plant biomass into high-quality organic biofertilizer [10].

The Australian silver wattle *Acacia dealbata* Link is one of the most prolific invasive plant species in Europe, threatening native habitats and biodiversity, thus being of great concern [11]. This species is able to invade intensive agricultural areas due to its active dispersal by several animals, water and wind, long-term soil-stored seed banks, and ability to prosper in low-nutrient substrates (reviewed in [11]). Although in Europe some laws prohibit its planting, its presence continues to increase, reducing native biodiversity by competing for resources with the native vegetation [11]. Given the great ecological threat *Acacia dealbata* represents, it is imperative to find alternatives for its sustainable control and mitigation. A proposed solution is the massive pruning of the species and the use of the generated biomass for bioactive compounds (e.g., [12]) or green manures (e.g., [13]).

The effects of the silver wattle *Acacia dealbata* composts on soil properties has been previously studied [14]; however, the characterization of the bacterial communities during vermicomposting of this species has never been addressed. In order to describe and understand the structure and role of the bacteriota during vermicomposting of silver wattle (*Acacia dealbata*), we coupled 16S rRNA high-throughput sequencing with metataxonomic analysis. Here, we characterized the microbial composition, diversity, and predicted metabolic function of the bacterial communities during vermicomposting of silver wattle for 56 days.

## 2. Material and Methods

### 2.1. Silver Wattle (Acacia Dealbata)

We manually collected young branches, leaves, and flowers of silver wattle trees (*Acacia dealbata* Link) in a forest near the University of Vigo in spring, when the trees were flowering. We chopped the branches into smaller sizes of 3–6 cm and left flowers and leaves intact, totaling 120 kg of fresh weight. Several physicochemical properties of the vermicomposting substrate were measured initially (day 0) and after 14, 28, 42, and 56 days: pH was 5.1 ± 0.04, 7.3 ± 0.07, 7.1 ± 0.1, 6.6 ± 0.2 and 6.4 ± 0.07; humidity was 64 ± 0.9%, 72 ± 4.9%, 79 ± 0.7%, 81 ± 0.4%, and 80 ± 0.6%; conductivity was 752 ± 35, 371 ± 23, 259 ± 18, 264 ± 14, and 212 ± 10 (mS cm^−2^); and nitrate (NO_3_) was 66 ± 56, 304 ± 297, 889 ± 270, 931 ± 108, and 872 ± 65 (µg N/g), respectively.

### 2.2. Vermicomposting Setup and Sampling Design

Vermicomposting of the silver wattle was performed in a rectangular metal pilot-scale vermireactor (4 m long × 1.5 m wide × 1 m high) housed in a greenhouse without temperature control. Before adding the silver wattle, the vermireactor contained a layer of vermicompost (12 cm height), which acts as a bed for the earthworms (*Eisenia andrei)*. Initially, earthworm population density in the vermireactor was 344 ± 10 individuals m^−2^ including 160 ± 10 mature earthworms m^−2^, 184 ± 9 immatures m^−2^, and 45 ± 8 cocoons m^−2^, with a mean biomass of 99.4 ± 4.5 g m^−2^. The silver wattle material was added to the vermireactor on top of a plastic mesh (5 mm mesh size) in a 12 cm layer. This plastic mesh is used to allow earthworm migration, prevent mixing of the processed silver wattle and the vermicompost bedding and simplify the silver wattle sampling during vermicomposting. We determined the density and biomass of the earthworm population periodically by collecting 10 samples (five from above and five from below the plastic mesh) of the material in the vermireactor with a core sampler (7.5 cm diameter and 12 cm height). The sampling was conducted every 14 days during the trial of the vermicomposting process, ending after 56 days and leaving 21 kg (fresh weight) of vermicompost.

Characterization of the microbial properties was conducted by dividing the silver wattle layer into five sections, and taking two samples (10 g) at random from each section at the beginning of the experiment (day 0) and after 14, 28, 42, and 56 days of vermicomposting. The two samples from each section were bulked and stored in plastic bags at −80 °C until analysis.

### 2.3. Microbial Activity

CO_2_ evolution was measured to determine basal respiration. Silver wattle samples (5 g fresh weight) were placed in 100 mL airtight glass vessels and incubated at 22 °C for 6 h. The CO_2_ produced from the sample was trapped in NaOH 0.02 M and measured subsequently by titration, with 0.01 M HCl to a phenolphthalein endpoint, after adding excess BaCl_2_.

### 2.4. DNA Sequencing and Analysis

DNA was extracted from 0.25 g (fresh weight) of silver wattle using the MO-BIO PowerSoil^®^ Kit following the manufacturer’s instructions. DNA quality and quantity were determined using BioTek’s Take3™ Multi-Volume Plate. To prevent contamination of the samples with microorganisms from the surrounding environment, all laboratory procedures were performed under a laminar flow hood. Sequencing the V4 hypervariable region of the 16S rRNA gene (~250 bp) was performed to assess bacterial succession by applying the dual-index sequencing strategy described by Kozich et al. [15]. In total, 25 DNA samples representing different sampling times (0, 14, 28, 42, and 56 days) were sequenced using the Illumina MiSeq platform at the Center for Microbial System, University of Michigan.

DADA2 (version 1.16) was used to infer the amplicon sequence variants (ASVs) present in each sample [16]. Exact sequence variants provide a more accurate and reproducible description of amplicon-sequenced communities than is possible with OTUs defined at a constant level (97% or other) of sequence similarity [17]. Bioinformatics processing followed the DADA2 pipeline tutorial (https://benjjneb.github.io/dada2/tutorial.html, accessed 1 March 2021). Forward/reverse read pairs were trimmed and filtered, with forward reads truncated at 220 nt and reverse reads at 130 nt, no ambiguous bases allowed, and each read required less than two expected errors based on their quality scores. ASVs were independently inferred from the forward and reverse of each sample using the run-specific error rates, and then read pairs were merged. Chimeras were identified in each sample and removed if identified in a sufficient fraction of the samples in which they were present. Taxonomic assignment was performed against the Silva v138 database using the implementation of the RDP naive Bayesian classifier available in the DADA2 R package (min boot 80) [18,19]. A total of 587,828 sequences (mean: 22,340; SD: 6052) passed all quality filters and were assigned to 3373 ASVs. Normalization of the number of sequences was performed by subsampling all samples to 20,267 sequences per sample. Rarefaction curves indicated that the sampling depth was optimal for all samples (Appendix A).

We predicted the functional composition of the metagenomes using the Phylogenetic Investigation of Communities by Reconstruction of Unobserved States software package (PICRUSt2) [20]. We followed the procedure described at https://github.com/picrust/picrust2/wiki/Full-pipeline-script (accessed on 1 March 2021) using the filtered dataset. The weighted nearest sequenced taxon index (NSTI) for our samples was 0.15 ± 0.01 (mean ± S.E.), which is acceptable as the samples were not from well described and/or sampled environments [21].

### 2.5. Statistical Analysis

We analyzed and plotted all data using R version 4.0.3 and used the packages phyloseq [22], ggplot2 [23], and ggtree [24]. Previous to statistical analysis, we conducted a prevalent filtering on our data, keeping only ASVs present in at least 10% of the samples. By doing this, we removed 57% of the ASVs, but only 5% of the sequences (Appendix A). Rarefaction curves indicated that the sampling depth was optimal for all samples for both the full dataset (3373 ASVs and 587,828 sequences) and the filtered dataset (1446 ASVs and 558,506 sequences, Appendix A). We normalized ASV counts using the variance-stabilizing transformation for analysis that assumed homoscedasticity or could be influenced by unequal variances [25]. We used raw ASV counts when analyzing differential ASV abundances with negative binomial models [25,26].

The differential abundance of ASV and bacterial phyla, classes, and genera during vermicomposting of silver wattle were analyzed using negative binomial models as implemented in the package DESeq2 [25]. Differential abundances of ASVs and other bacterial taxa were determined according to Wald tests and *p*-values adjusted by false discovery rate (*p*-adj < 0.05). We further adjusted “raw” *p* values using the Benjamini–Hochberg method to correct for the multiple pairwise Wald tests conducted for each time-to-time comparison (0–14, 14–28, 28–42, and 42–56 days). After correction, non-significant contrasts were considered to have an effect size (log2 fold change) of zero. We followed the same procedure for the functional profiling using the raw output from PICRUSt2.

An approximately maximum-likelihood phylogenetic tree was inferred using FastTree 2.1 [27]. After removing ASVs from silver wattle samples (day 0), we defined the core microbiome of vermicomposting of the silver wattle as that comprised of ASVs present in all the samples processed by earthworms, that is, samples of 14, 28, 42, and 56 days.

Taxonomic α-diversity was calculated as the number of observed ASVs, and diversity and richness were estimated by the inverse Simpson and Chao1 indexes, respectively. Phylogenetic diversity was calculated using Faith’s phylogenetic diversity [28]. The effect of time (0, 14, 28, 42, and 56 days) on α-diversity of bacterial communities from the silver wattle during vermicomposting was analyzed using mixed models in the ‘nlme’ R package [29]. Time was set as the fixed factor and the effect of time nested in each sample was considered as a random factor to account for non-independence of samples due to repeated measures (response variable~time, random = ~1|subject/time). Additionally, the normality of residuals and homogeneity of variance across groups was checked for each variable. Tukey’s test was used for post-hoc comparisons, and Benjamini–Hochberg FDR was used as a multiple test correction method using the ‘multcomp’ package in R [30].

We estimated taxonomic β-diversity at the ASV level as the difference in the composition of the bacterial taxonomic community between samples from different times during vermicomposting. This was conducted by coupling principal coordinate analysis (PCoA) with distance matrices that take the abundance of ASVs into account (Bray-Curtis) or not (Jaccard). We also estimated phylogenetic β-diversity by PCoA of weighted (considering abundance of ASVs) and unweighted unifrac matrix distances [31] using the phyloseq library [22]. As before, we analyzed differences in β-diversity of bacterial communities from the silver wattle during vermicomposting using mixed models with PCoA scores as variables and time as fixed factor and the effect of time nested in each sample as a random factor. Tukey’s test and Benjamini–Hochberg FDR were also used for post-hoc comparisons and as the multiple test correction method, respectively.

For the functional profiling, mean values of relative abundance of selected gene contents were analyzed with mixed models and post-hoc test as described above.

## 3. Results

Below, we describe the microbial composition and structure, diversity, and predicted metabolic function of the bacteria during vermicomposting of silver wattle at five time points (0, 14, 28, 42, and 56 days).

### 3.1. Changes in Earthworm Density and Microbial Activity during Vermicomposting

Earthworm density significantly increased since the beginning of the trial until day 28, after which it significantly decreased between days 42 and 56 (*p* < 0.0001, Figure 1, inset). Microbial activity, measured as basal respiration, significantly decreased during vermicomposting until day 28 and then again between days 42 and 56 (*p* < 0.0001, Figure 1). Additionally, it was possible to see a high correlation between earthworm density and basal respiration (Figure 1).

### 3.2. Changes in Bacterial Community Composition during Vermicomposting

The highest number of significantly different taxa was observed between days 0 and 14 (16 phyla, 30 classes, and 177 genera; *p* < 0.05, Appendix A), except at the ASV level, which was observed between days 14 and 28 (150 ASVs; *p* < 0.05, Appendix A). Significant differences in bacterial composition decreased up to day 42, increasing again between days 42 and 56 (Appendix A).

Initial bacterial community composition (day 0) was dominated by the phylum Proteobacteria, class Gammaproteobacteria, and genus *Pseudomonas* (Figure 2). The relative abundance of these taxa significantly decreased between days 0 and 14 (*p* < 0.01, Appendix A; Figure 2). Nevertheless, Proteobacteria continued to dominate the bacterial microbiome along with Bacteroidota, which significantly increased its abundance after 14 days of vermicomposting (*p* < 0.01, Appendix A; Figure 2). The abundance of sequences belonging to Acidobacteriota, Actinobacteriota, Myxococcota, and Verrucomicrobiota was small but significantly increased during vermicomposting when compared to their relative abundance at day 0 (*p* < 0.04, Appendix A; Figure 2). Bacteroidia was the dominant class between days 14 and 56, while the relative abundance of the main genera varied (Figure 2).

### 3.3. Changes in α- and β-Diversity during Vermicomposting

Bacterial communities in fresh silver wattle (day 0) presented low α-diversity for all diversity indices (Figure 3A and Appendix A). After earthworms started vermicomposting, significant increases in α-diversity were observed between days 0 and 14 and days 14 and 28 (*p* < 0.0001), stabilizing after day 28 (Figure 3A and Appendix A). The only exception to this trend was observed for Faith’s phylogenetic diversity index, with no significant differences observed between days 14 and 28 and days 42 and 56 (Appendix A). Significant differences were also observed in phylogenetic and taxonomic β-diversity (*p* < 0.0001, Figure 3B and Appendix A). Along the first dimension of the principal coordinate analysis plot, the bacterial composition showed significant differences between fresh silver wattle (day 0) and vermicomposted silver wattle (days 14–56) for the weighted unifrac index (Figure 3B). The second dimension reflected the changes in bacterial community composition between all stages of the vermicomposting process (Figure 3B and Appendix A). Samples from days 28, 42, and 56 clustered together, while samples from days 0 and 14 grouped in their own cluster (Figure 3B and Appendix A).

### 3.4. Core Microbiome during Vermicomposting

Ten ASVs were identified as the bacterial core microbiome during vermicomposting of silver wattle, being present in all samples from days 14 to 56 (Figure 4). The fresh silver wattle (day 0) was not considered within the core microbiome since these samples were not processed by earthworms. Six of these ASVs belonged to the phylum Proteobacteria and the other four to the phylum Bacteroidota (Figure 4). The phylum Proteobacteria comprised ASVs from the families Comamonadaceae (ASV9) and Moraxellaceae (ASV30), and from the genera *Novosphingobium* (ASV83), *Dokdonella* (ASV52), *Achromobacter* (ASV35), and *Methylophylus* (ASV116). Among the Bacteroidota present, one ASV belonged to the family Microscillaceae (ASV49) and the remaining to the genera *Flavobacterium* (ASV3 and ASV6) and *Ohtaekwangia* (ASV31). The abundance of all 10 ASVs significantly changed between days 0 and 14 (Appendix A). Additionally, the abundance of ASV6 (*Flavobacterium*) differed between days 14 and 28 (Appendix A).

### 3.5. Functional Diversity during Vermicomposting

Metagenomic predictions using PICRUSt2 showed distinct profiles of certain functional genes such as those involved in cellulose metabolism and nitrification and salicylic acid for different days of vermicomposting (Figure 5). The relative abundance of genes related to cellulose metabolism significantly increased between days 14 and 28 and decreased later between days 42 and 56, with no significant differences between samples from days 0, 14, and 56 (Figure 5A). Genes related to nitrification increased their relative abundance since the beginning up to day 28 of vermicomposting, after which they stabilized (Figure 5B). Genes involved in the synthesis of salicylic acid significantly increased during days 14 and 28, with no significant difference between these two time points, significantly reducing their abundance at day 42 and again at day 56 (Figure 5C).

## 4. Discussion

The present study provides insights regarding the vermicomposting of the silver wattle *Acacia dealbata* from a microbial perspective. Vermicomposting of the silver wattle followed the normal pattern of accelerated decomposition observed in previous studies using vegetable wastes (e.g., [30]) and characterized by a rapid reduction in microbial activity. These results indicate the success and good performance of the vermicomposting process.

Our results showed that bacterial community composition was driven by changes in the organic source during vermicomposting, being classified in three groups. The first group is represented by the microbial community of the fresh silver wattle that has not been through the earthworm gut (day 0). This group is characterized by low and distinct microbial diversity, mainly dominated by Proteobacteria, as seen in previous studies [10,32]. This stage has also been characterized by its low microbial metabolic activity (e.g., [10,32,33]). Accordingly, our results showed significantly lower gene counts of predicted functions related to nitrification and salicylic acid synthesis when compared to the remaining days of vermicomposting.

At day 14, the second group was comprised by earthworm-gut bacteria. Significant changes in microbial composition were defined by significant increases in microbial diversity and predicted metabolism. At this stage, a turnover of the microbial community had occurred, with the most notable changes being the significant increase in Bacteroidota and a significant decrease in Proteobacteria, consistent with previous findings [10,34]. Additionally, the abundance of copiotroph taxa gave place to oligotrophs between days 0 and 14. Copiotrophic bacteria, characterized by higher rates of carbon turnover, are commonly found during the initial stages of vermicomposting due to the fact that there is a fast decomposition of microorganisms and nutrients that just pass through the gut of earthworms, producing labile nutrients [6,35]. Oligotrophic bacteria will then take over as microbial succession moves forward, showing a higher ability to efficiently metabolize the remaining recalcitrant substrate during vermicomposting [6,35]. On the other hand, phylogenetic diversity, as denoted by Faith’s PD, was not significantly different at this point, indicating close phylogenetic relatedness of microbial species along the bacterial succession observed during vermicomposting.

The third group, here represented by days 28, 42, and 56, resulted from gradual changes in the quality and quantity of the available nutrients occurring with microbial succession. The microbial communities during these time points were overall similar in diversity, although some changes in composition and predicted metabolism were observed. A peak in diversity during the last stage of vermicomposting had also been previously observed (e.g., 42–91 days [10]; 30–45 days [34]).

The increase in the predicted bacterial function during vermicomposting of the silver wattle observed here was in line with previous observations of improved plant performance when vermicompost is used as a soil additive (e.g., [10,33]). For example, resistance mechanisms against plant pathogens and enhanced plant growth and performance have been shown to be induced by salicylic acid (e.g., [36,37,38]). Additionally, cellulose degradation is an important component of the vermicomposting process of plant materials, and seemed to increase here as well as in other studies (e.g., [10,33]). Increased nitrification is also expected during vermicomposting, being previously considered as an indicator of mature compost [39]. Additionally, a nitrification increase was also confirmed through measurements of nitrate in the vermicompost substrate; nitrate (NO_3_) concentrations augmented since day 0 (66 ± 56 µg N/g) until day 56 (872 ± 65 µg N/g). Some of the observed changes in the predicted microbial metabolism might be related to changes in the community composition at the phylum level, likely to be involved in the breakdown of the silver wattle. For example, Actinobacteriota and Bacteroidota were previously seen to be involved in cellulose degradation (e.g., [40,41]). Changes to the functional diversity of the microbiome illustrate possible mechanisms by which plant performance is improved when using vermicompost as a plant biostimulant.

The vermicomposted samples from silver wattle between days 14 and 56 showed 10 core ASVs (i.e., ASVs present in all samples). Some of these ASVs belonged to genera usually found in compost and vermicompost such as *Dokdonella* (e.g., potting soil [42]) or *Flavobacterium* (e.g., sewage sludge and cattle dung [43]). Moreover, *Flavobacterium* has been shown to be highly effective at enhancing sludge stabilization during the vermifiltration process of excess sewage sludge [44]. It has also been reported to enhance the biodegradation of the contaminant pentachlorophenol in soil [45], which suggests that it can be stimulated during vermicomposting [46]. Denitrification by bacteria is one of the most important processes through which fixed nitrogen is returned from the soil to the atmosphere [47]. The genus *Methylophylus* and species from the family Comamonadaceae, here part of the core microbiome, were previously associated with denitrification activity of the soil (e.g., [48,49]). Another core ASVs belonged to the genus *Novosphingobium*, which has been associated with the degradation of xenobiotic compounds [50]; this suggests the presence of pollutants in silver wattle, likely as a result of industrial activity [51]. Additionally, this genus has been shown to alleviate salt stress and promote plant growth of citrus plants (e.g., [52]), reinforcing the potential of silver wattle vermicompost as a fertilizer. Finally, core ASVs belonging to the genus *Achromobacter* found here have been partially related with pesticide degradation during vermicomposting [53].

To conclude, we provide a detailed picture of the bacterial succession during the vermicomposting of the silver wattle *Acacia dealbata*. We show that the vermicompost of this species generates a final product with high microbial diversity and function. This study adds to an increasing body of literature showing the benefits of vermicomposting to degrade invasive plant species as well as sustainable use of the final vermicompost as a plant growth catalyst and disease suppressor.

## Figures and Tables

**Figure 1 microorganisms-10-00065-f001:**
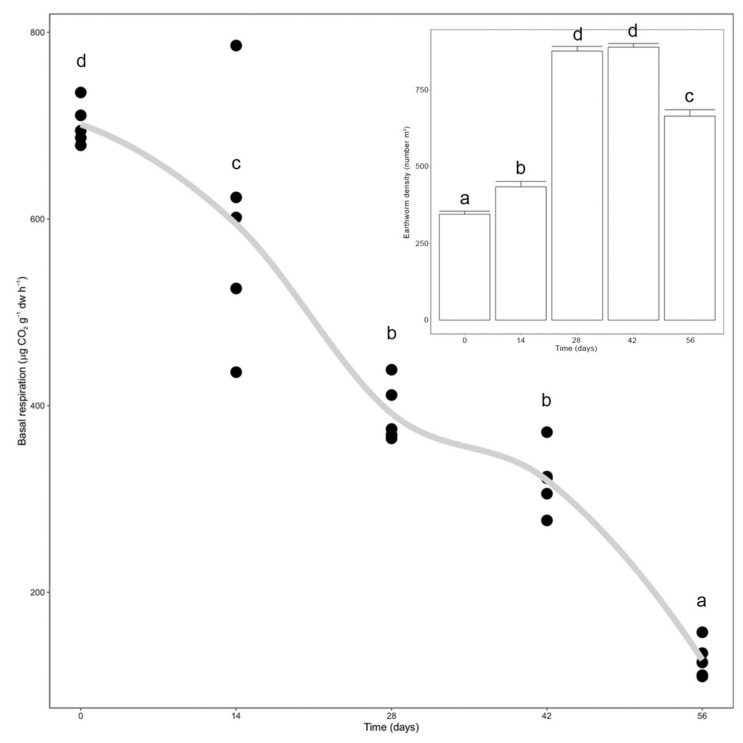
Variation in microbial respiration and earthworm density (inlet) during vermicomposting of the silver wattle. Individual values (*n* = 5) were plotted for each time point, and the curve was plotted using the “loess” smoothing method in ggplot2. Earthworm biomass values are presented as means ± standard error (*n* = 5). Letters indicate significant differences between time points.

**Figure 2 microorganisms-10-00065-f002:**
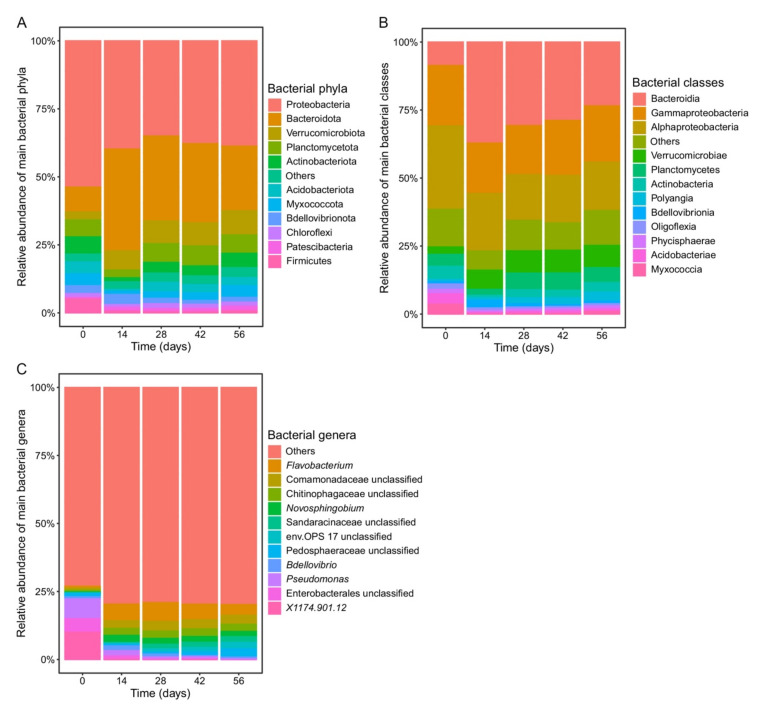
Changes in the bacterial community composition at the phylum (**A**), class (**B**), and genus (**C**) level during vermicomposting of silver wattle. Bars represent the relative abundance of dominant bacterial phyla. Low abundant bacterial taxonomies (relative abundance < 1%) were collapsed into “Others”.

**Figure 3 microorganisms-10-00065-f003:**
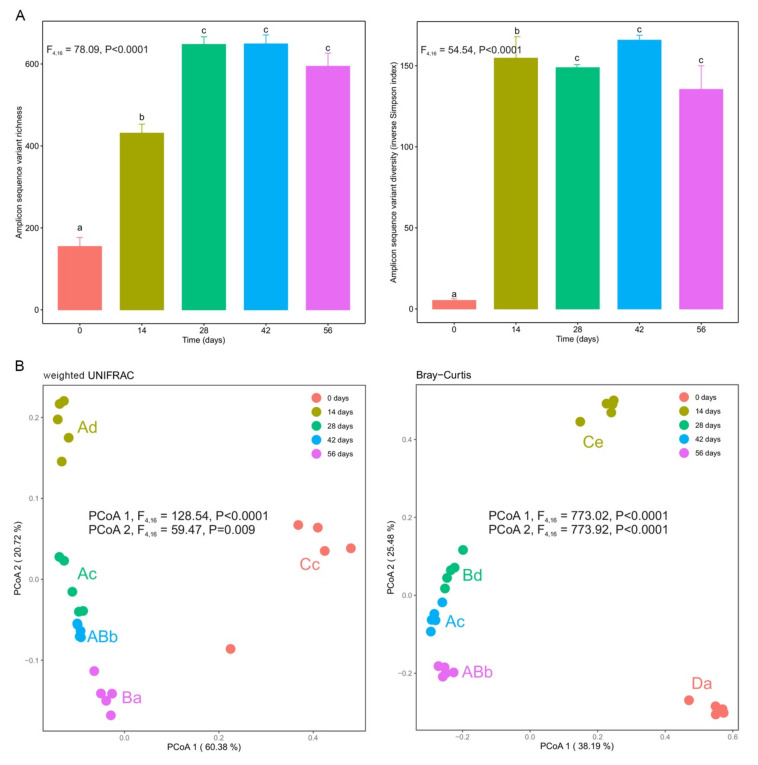
Changes in bacterial α- and β-diversity during vermicomposting of the silver wattle. (**A**) α-diversity is shown in terms of amplicon sequence variant (ASV) taxonomic richness and diversity (inverse Simpson index). Letters indicate significant differences between time points (Tukey HSD test). (**B**) β-diversity is shown with principal coordinate analysis (PCoA) of weighted UniFrac and Bray–Curtis distances. Capital and lower case letters indicate significant differences between the time points in PCoA1 and PCoA2 scores, respectively (Tukey HSD test, FDR corrected).

**Figure 4 microorganisms-10-00065-f004:**
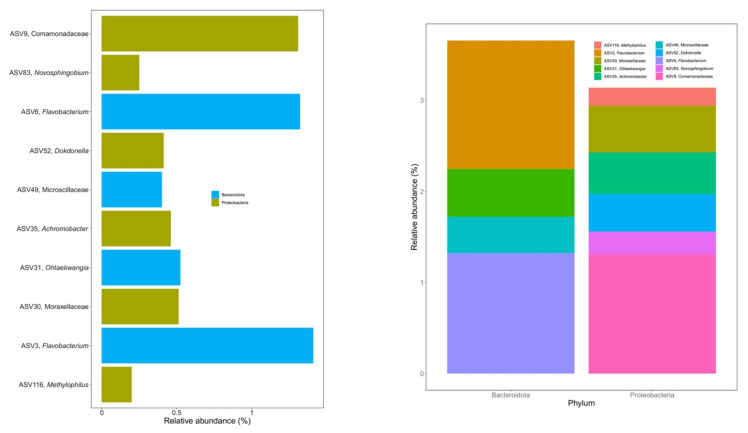
Core microbiome of the silver wattle vermicompost. Initial substrate (day 0) was not considered for determination of the core microbiome. The listed 10 ASVs represent 8.56% of sequences from days 14 to 56 and were found in all samples (*n* = 20).

**Figure 5 microorganisms-10-00065-f005:**
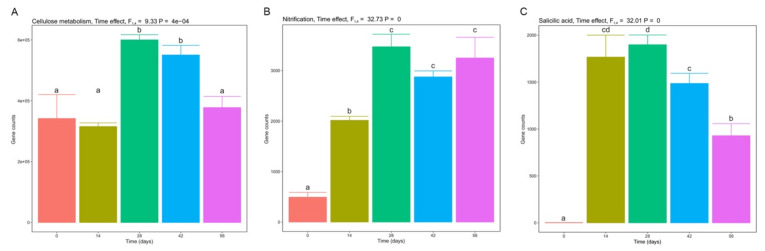
Changes in gene abundance of PICRUSt-predicted KEGG orthologies implied in cellulose metabolism (**A**), nitrification (**B**), and salicylic acid synthesis (**C**) during vermicomposting of the silver wattle. Values are presented as means ± standard error (*n* = 5). Above each plot, results from mixed-effects models are shown. Letters indicate significant differences between time points (Tukey HSD test).

## Data Availability

Sequence data have been uploaded to the GenBank SRA database under accession PRJNA772065.

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
