# Peer review of "Bacterial Succession during Vermicomposting of Silver Wattle (Acacia dealbata Link)"

_microorganisms, 2021, doi:10.3390/microorganisms10010065_

Round 1
Reviewer 1 Report
The manuscript is a description of the changes in soil microbial community of a worm based compost when supplemented with silver wattle. The investigation looked at the changes in earthworm density, CO2 production and the microbial community structure over 56 days. The study is interesting and in particular the introduction and methods provide a clear rationale and reproducible methods set which are very well written. I have minor comments with regards to the figures which I believe would improve the impact of the manuscript.
Presentation of figures. The order of the bars in the taxa bar charts in figure 3 should be changed such that rather than A-Z order the bars are ordered in something such as sum total order, mainly so that the major contributors can be easily compared in terms of % abundance. I would also like to see the other category have a capital letter. In my version of the manuscript, the colour palette has not copied across particularly well and has 'striations' in some of the bars. In the previous figure 2, it is clear that there are 5 different communities, why is the variation in these communities not further considered in figure 3? Potential scope for line graphs of the main changes to the community structure over time (with error bars) to demonstrate changes (e.g G- proteobacteria/bacteroidia shift). This would then potentially lead on to the discussion of ASVs in figure 5
Considering that the number of taxa observed at the Genus level in a >1% abundance is quite low, is there much merit in figures 3B and C when A shows the overall picture and D shows specific genus level changes quite well?
Figure 4, in the PCoA, could you have added CO2/earthworm density as environmental factors as vectors?
Figure 6- what was the reasoning behind the selection of nitrification and salicylic acid gene pathways? was this purely on gene counts compared to other ontologies? also salicylic acid is spelled differently in the figure and caption.
Within the conclusion, although adequately referenced, I do think that there is some consideration needed to bring together all the data points. For instance, there is discussion of cellulose, nitrification and sialcylic acid GO, but these are not really linked to the ASVs that are identified, there is discussion of the organisms, but not whether or not these are responsible for the specific counts within Figure 6.
In terms of the GO's, could there be scope for measurement in nitrification (nitrate loss), sialcylic acid from the soil to confirm these observations?
Author Response
The manuscript is a description of the changes in soil microbial community of a worm based compost when supplemented with silver wattle. The investigation looked at the changes in earthworm density, CO2 production and the microbial community structure over 56 days. The study is interesting and in particular the introduction and methods provide a clear rationale and reproducible methods set which are very well written. I have minor comments with regards to the figures which I believe would improve the impact of the manuscript.
Presentation of figures. The order of the bars in the taxa bar charts in figure 3 should be changed such that rather than A-Z order the bars are ordered in something such as sum total order, mainly so that the major contributors can be easily compared in terms of % abundance. I would also like to see the other category have a capital letter. In my version of the manuscript, the colour palette has not copied across particularly well and has 'striations' in some of the bars. In the previous figure 2, it is clear that there are 5 different communities, why is the variation in these communities not further considered in figure 3? Potential scope for line graphs of the main changes to the community structure over time (with error bars) to demonstrate changes (e.g G- proteobacteria/bacteroidia shift). This would then potentially lead on to the discussion of ASVs in figure 5.
A: We thank the reviewer for these suggestions. We have collapsed figure 2 and 3 together (new figure 2) and changed the order of the bars to be presented in sum total order and the other category to have a capital letter. Regarding the addition of line graphs, we decided not to add them because we have other figures already showing taxonomic shifts.
Considering that the number of taxa observed at the Genus level in a >1% abundance is quite low, is there much merit in figures 3B and C when A shows the overall picture and D shows specific genus level changes quite well?
A: The reviewer is right and so we deleted panels B and C from figure 3 and have collapsed it with figure 2 (new figure 2).
Figure 4, in the PCoA, could you have added CO2/earthworm density as environmental factors as vectors?
A: Earthworm density and CO2 are highly correlated (Figure 1 and figure below); hence we didn’t include it in figure 4 (new figure 3). We have also added to the “Results” section: “Additionally, it is possible to see a high correlation between earthworm density and basal respiration (Figure 1).”.
Figure 6- what was the reasoning behind the selection of nitrification and salicylic acid gene pathways? was this purely on gene counts compared to other ontologies? also salicylic acid is spelled differently in the figure and caption.
A: We thank the reviewer for this comment. We have selected pathways that presented high gene counts and showed significant differences throughout the vermicomposting process. Additionally, among all possible KOs and metabolic pathways, we chose those representatives of biological properties of vermicompost, i.e., degradation of complex molecules, mineralization and promoting plant growth. We have now corrected the spelling of “salicylic” in figure 6 (new figure 5).
Within the conclusion, although adequately referenced, I do think that there is some consideration needed to bring together all the data points. For instance, there is discussion of cellulose, nitrification and sialcylic acid GO, but these are not really linked to the ASVs that are identified, there is discussion of the organisms, but not whether or not these are responsible for the specific counts within Figure 6.
A: We thank the reviewer for the comment however, it is not possible to ascertain for sure that the organisms identified are indeed responsible for specific GO counts. Even though it is possible to analyze taxon function contribution, we decided not to add this analysis, since picrust2 results are limited by the currently available genomes and biased towards human health microorganisms. Based on available literature, we have speculated and discussed the relationship between organisms and GO’s as much as possible.
In terms of the GO's, could there be scope for measurement in nitrification (nitrate loss), sialcylic acid from the soil to confirm these observations?
A: We thank the reviewer for this suggestion. Measurements of salicylic acid were not conducted in the scope of this study; however, nitrate was measured throughout vermicomposting. This information was added to “Material and Methods” section, “Silver wattle (Acacia dealbata)” subsection (“Several physicochemical properties of the vermicomposting substrate were measured initially (day 0) and after 14, 28, 42 and 56 days: (…) nitrate (NO3) was 66±56, 304±297, 889±270, 931±108 and 872±65 (µg N/g), respectively.”), as well as to the “Discussion” (“Additionally, a nitrification increase was also confirmed through measurements of nitrate in the vermicompost substrate; nitrate (NO3) concentrations augmented since day 0 (66±56 µg N/g) until day 56 (872±65 µg N/g)”).

Reviewer 2 Report
L22, use the full name to replace the ASVs.
L71-73, please provide the properties of vermicomposting substrate.
L192-193, please show some data about the changes of physicochemical properties during vermicomposting, because it affects the micrbial community directly.
For all figures, please enlarge the font numbers in all coordinate axises.
Author Response
L22, use the full name to replace the ASVs.
A: We added “Amplicon Sequence Variants” to the sentence.
L71-73, please provide the properties of vermicomposting substrate.
A: We thank the reviewer for this suggestion and added the properties of vermicomposting substrate to the “Material and Methods” section, “Silver wattle (Acacia dealbata)” subsection (“Several physicochemical properties of the vermicomposting substrate were measured initially (day 0) and after 14, 28, 42 and 56 days: pH was 5.1±0.04, 7.3±0.07, 7.1±0.1, 6.6±0.2 and 6.4±0.07; humidity was 64±0.9%, 72±4.9%, 79±0.7%, 81±0.4% and 80±0.6%; conductivity was 752±35, 371±23, 259±18, 264±14 and 212±10 (mS cm−2); and nitrate (NO3) was 66±56, 304±297, 889±270, 931±108 and 872±65 (µg N/g), respectively.”).
L192-193, please show some data about the changes of physicochemical properties during vermicomposting, because it affects the microbial community directly.
A: Thank you for the suggestion. We added the measured physicochemical properties during vermicomposting to the “Material and Methods” section, “Silver wattle (Acacia dealbata)” subsection (see previous comment).
For all figures, please enlarge the font numbers in all coordinate axes.
A: We have enlarged the font in all figures.
Reviewer 3 Report
The paper is well written and designed. The results are important for the readres. Congratulations.
Author Response
The paper is well written and designed. The results are important for the readres. Congratulations.
A: We thank the reviewer for the positive comment.